# Acrylamide Content of Experimental Flatbreads Prepared from Potato, Quinoa, and Wheat Flours with Added Fruit and Vegetable Peels and Mushroom Powders

**DOI:** 10.3390/foods8070228

**Published:** 2019-06-26

**Authors:** Lauren M. Crawford, Talwinder S. Kahlon, Selina C. Wang, Mendel Friedman

**Affiliations:** 1Department of Food Science and Technology, University of California, Davis, CA 95616, USA; 2Western Regional Research Center, Agricultural Research Service, U.S. Department of Agriculture, Albany, CA 94710, USA

**Keywords:** potato flatbreads, quinoa flatbreads, wheat flatbreads, acrylamide, food safety, health benefits

## Abstract

Flatbreads are a major food consumed worldwide. To mitigate an undesirable safety aspect of flatbreads that might be associated with the potentially-toxic compound acrylamide, we recently developed recipes using a variety of grains that resulted in the production of low-acrylamide flatbreads. To further enhance the functionality of flatbreads, we have developed, in this work, new experimental flatbreads using potato, quinoa, and wheat flours supplemented with peel powders prepared from commercial nonorganic and organic fruits and vegetables (apples, cherry tomatoes, melons, oranges, pepino melons, sweet potato yams), potato peels, and mushroom powders (Lion’s Mane, *Hericium erinaceus*; Reishi, *Ganoderma lucidum*; and Turkey Tail, *Trametes versicolor*). These additives have all been reported to contain beneficial compositional and health properties. The results of fortification of the baked flatbreads showed either no effect or increases in acrylamide content by unknown mechanisms. Since the additives did not increase the acrylamide content of the quinoa flour flatbreads for the most part, such supplemented quinoa flatbreads have the potential to serve as a nutritional, gluten-free, low-acrylamide, health-promoting functional food. Mushroom powder-fortified wheat flatbreads with relatively low acrylamide content may also have health benefits.

## 1. Introduction

Acrylamide (CH_2_ = CH–CONH_2_) is an industrially produced conjugate molecule that is also formed during thermal processing of plant-based foods such as cereals, potatoes, coffee, and olives—all of which are widely consumed [1]. The primary mechanism for acrylamide formation has been extensively reviewed and involves a reaction between the free amino acid asparagine and the reducing sugars glucose and fructose [2,3,4]. After consumption, acrylamide and the in vivo-formed acrylamide epoxide (glycidamide) have been shown to act—by nucleophilic addition reaction—as biological alkylating agents, forming adducts with the SH group of L-cysteine and glutathione, essential structural proteins and enzymes, the NH_2_ groups of hemoglobin, and DNA of the structural type R–X–CH_2_–CH_2_–CONH_2_ and R–X–(CHOH)–CH_2_–CONH_2_, where X = S or NH. These in vivo reactions seem to be responsible for the reported adverse effect in animals and possibly humans, including carcinogenicity, neurotoxicity, teratogenicity, and anti-fertility effects [3,5,6,7]. Since its discovery as a food processing byproduct about 20 years ago, ongoing worldwide studies have been designed to (a) discover low-acrylamide foods; (b) reduce the acrylamide content of foods during industrial and home processing (baking, frying); and (c) inhibit adverse effects in vivo. Guidelines recommended by government agencies for the maximum content of acrylamide in foods and the estimated consumption of acrylamide by different populations in 14 countries have been described in our previous study [8].

As part of an effort to develop health-promoting flatbreads with low acrylamide content, we previously reported that flatbreads made using flours from ancient grains such as brown rice, buckwheat, cornmeal, millet, oats, and quinoa had a lower acrylamide content than flatbreads prepared from potato and wheat flours [8]. We are also engaged in a parallel effort designed to define the potential health benefits of antioxidative fruits and vegetable peels, as well the peel (hull) from rice, which are often discarded as a food-processing waste product [9,10,11]. As part of this effort, we have shown that (a) the dietary supplementation of potato peel powders reduced weight gain in mice on a high-fat diet [12]; (b) potato peel powders inhibited the growth of pathogenic trichomonads (protozoa) that cause sexually transmitted diseases in humans and cattle [13]; (c) apple, orange, and potato peel powders inhibited the outgrowth of pathogenic *Bacillus cereus* bacteria from spores in cooked rice [14]; and (d) a rice hull smoke extract reduced weight gain in mice on a high-fat diet [15]. We also recently determined the content of phenolic and flavonoid compounds and antioxidative activities of 12 melon peel powders [16].

These considerations, as well as published studies on the dual-inhibiting and -enhancing effects of plant polyphenols on acrylamide formation and elimination [17,18,19,20,21], led us to study the acrylamide content of flatbreads prepared from potato, quinoa, and wheat flours supplemented with 5% and 10% fruit and vegetable peels and mushroom powders. The peel powders were prepared from commercial fruits and vegetables, which are rich sources of antioxidative and anti-inflammatory compounds. Our ultimate objective is to create healthy gluten-free flatbreads that have high nutritional value, added health benefits, and a low acrylamide content.

## 2. Materials and Methods

### 2.1. Materials

The following tests substances were purchased at local markets (Berkeley Bowl, Costco, Safeway, Whole Foods): potato, quinoa, and whole-wheat flours; Red Delicious and Fuji nonorganic and organic apples; Mandarin, Valencia, and navel oranges; Garnet and purple sweet potato yams; Galia organic and nonorganic melons; pepino melons; and red cherry tomatoes. Umatilla Russet frozen potato peels, a byproduct of the industrial preparation of French fries, were a gift from Dr. Jeff Bohlscheidt of the Simplot Company (Caldwell, ID, USA). Lion’s Mane, Reishi, and Turkey Tail mushroom powders of unknown origin were obtained from an online Healing Spirits Herb Farm and Education Center (Avoca, NY, USA). Acrylamide (≥99% purity) and acrylamide-d3 solution (500 mg/L in acetonitrile) were purchased from Millipore Sigma (St. Louis, MO, USA). Optima LC/MS grade acetonitrile, methanol, and formic acid, and HPLC grade hexane were purchased from Fisher Scientific (Fair Lawn, NJ, USA).

### 2.2. Preparation of Fruit and Vegetable Peel Powders

As previously reported [12,14], whole fruits and vegetables, except cherry tomatoes, were individually washed with water, dried with absorbent paper tissue, and manually peeled using a potato peeler. The cherry tomatoes were subjected to blanching before peeling. After manual removal of the stems, tomatoes were washed with water using a brush. An X shape was then cut into the bottom of each tomato using a knife. These were then placed into a saucepan with boiling water for 60 s. The tomatoes were scooped up with a large spoon and placed into a container with ice water. Loose peels of the cooled, wrinkled, blanched tomatoes were then manually removed. All wet peels were freeze-dried, and the resulting dry peels were ground into fine powders using an electric coffee grinder (Kreps, Millville, NJ, USA). The final yields of the peel powders, approximately 10–13%, were determined from the original weights of the fruits and vegetables and those of the freeze-dried peel powders. For example, the powder from 20.5 g of organic red grape tomatoes weighed 2.7 g, which corresponds to a yield of 12.6%.

### 2.3. Preparation of Peel- and Mushroom-Fortified Flatbreads

Finely ground potato flour from 100% dehydrated whole potatoes and whole-wheat flour were obtained from Bob’s Red Mill (Milwaukie, OR, USA). Kirkland Signature quinoa grains were obtained from Costco. The experimental flatbreads were prepared in our laboratory as previously described [22]. Water was added to the flatbread ingredients until the dough began forming a ball. The dough was kneaded until it became smooth and elastic, placed in a 200 × 100 No. 3180 Pyrex bowl, covered with a polyvinyl film (Polyvinyl Films, Sutton, MA, USA), and kept at room temperature for 30 min. Small portions (50 g) of dough were put on parchment paper (nonstick, oven-safe up to 216 °C) and pressed into about a 17 cm circle in a 20 cm Alpine Cuisine flatbread maker (Aramco Imports, Inc., Commerce, CA, USA) to a thickness of about 1–1.5 mm. Flatbreads were cooked between upper and lower hot irons of the flatbread maker for 2.5 min (1.25 min each side) for potato and quinoa, and 2.0 min (1 min each side) for whole wheat at 195.5 °C on parchment paper in a 1000 Watt CucinaPro Flatbread Maker (SCS Direct, Inc., Trumbull, CT, USA). Cooked flatbreads were chopped for 30 s in a Cuisinart coffee grinder Model DCG-20N (Cuisinart East Windsor, NJ, USA). Chopped flatbreads were dried at 103 °C for 3 h (AOAC 935.29). For the statistical analysis of reproducibility, three separate dry flatbreads were individually ground into fine powders using Cuisinart coffee grinder. The compositions of potato, quinoa, and wheat flour flatbreads with various fruit and vegetable peels and mushroom powders at a concentration of 5% and 10% are shown in Table 1, Table 2 and Table 3.

### 2.4. Analysis of Acrylamide by HPLC–Tandem Mass Spectrometry

Acrylamide in flatbreads was extracted and analyzed as previously described [8]. Briefly, flatbread powder (2.00 ± 0.01 g) was extracted with 19.5 mL of methanol/water 80:20 (v/v). Acrylamide-d3 solution (0.5 mL, 400 µg/L in water) was added, and the sample was stirred for 20 min, followed by centrifugation (2700× *g*, 5 min). The supernatant was mixed with 10 mL of hexane, shaken for 45 s, and centrifuged again (1000× *g*, 3 min). The hexane layer was removed. Carrez reagents I and II (0.1 mL) were added, and the sample was shaken for 30 s. After a final centrifugation (2700× *g*, 5 min), 10 mL of extract was evaporated to dryness at 35 °C using a rotary evaporator. Nanopure water (1 mL) was used to redissolve the residue. The sample was filtered (0.45 µm, nylon) and stored at 4 °C until analysis (within two days).

Acrylamide analysis was performed using a Waters Alliance 2695 Separation Module coupled to a Waters Quattro Micro API Mass Spectrometer (Waters Corporation, Milford, MA, USA) operating in positive electrospray ionization mode. An Agilent C18 Eclipse Plus column (5 µm, 4.6 mm × 250 mm) was used for separation. Chromatographic and mass spectrometer conditions were identical to those used in the previous publication [8]. Acrylamide-d3 and acrylamide had retention times of 9.1 and 9.2 min, respectively. Acrylamide was quantified using the m/z 72 → m/z 54.8 transition and acrylamide-d3 was quantified using the m/z 75 → m/z 57.8 transition. Acrylamide-d3 was added to standard solutions of acrylamide at the same final concentration (100 µg/L) as the samples. Six-point calibration curves were constructed by plotting concentration versus the peak area ratio of acrylamide to acrylamide-d3. The range of the curve varied for each type of flatbread based on sample concentrations: quinoa, 0–102 µg/L (R^2^ = 0.9998); potato, 0–1505 µg/L (R^2^ = 0.9999); whole wheat, 0–243 µg/L (R^2^ = 0.9999). The limit of detection and limit of quantification were determined as 3- and 10-times the signal-to-noise ratio, respectively, of acrylamide in the quinoa samples.

### 2.5. Statistics

Samples passed the Shapiro–Wilk test for normality and the Brown–Forsythe test for homoscedasticity. As a result, we used the two-tailed independent samples t-test to compare each treatment flatbread with the control to determine whether the additive had a significant effect on the acrylamide level (*p* ≤ 0.05).

## 3. Results

### 3.1. Precision, Sensitivity, and Reproducibility of the Acrylamide Analytical Method

Each type of flatbread was prepared in triplicate, and one analysis was performed on each replicate flatbread to obtain the acrylamide values. Ten individual flatbread replicates were also measured in duplicate to verify the precision of the analytical method. The relative standard deviation between duplicate measurements was <10% for all samples, ranging from 0.2–8.8%. The limit of detection was 1.8 µg/kg (1.8 parts per billion) and the limit of quantification was 6 µg/kg (6 parts per billion). The acrylamide levels of the three unfortified samples in this study were very consistent with those reported in our previous work: wheat, 21.3 µg/kg and 29.4 µg/kg respectively; quinoa, 6.6 µg/kg and 6.4 µg/kg respectively; potato, 470 µg/kg and 487 µg/kg respectively [8]. It is particularly noteworthy that the presence of multiple food-based additives that contain many structurally different ingredients, as mentioned below, did not interfere with the acrylamide analysis.

### 3.2. Analysis of Acrylamide in Potato, Quinoa, and Whole-Wheat Flatbreads

Figure 1 depicts the retention times of HPLC-MS/MS chromatograms of the internal standard (acrylamide-d3) and of acrylamide in (A) 100 µg/L standard; (B) quinoa flatbread; (C) potato flatbread; and (D) whole-wheat flatbread. Table 1, Table 2 and Table 3 show the composition of the doughs used to prepare the potato, quinoa, and wheat flatbreads. Table 4, Table 5 and Table 6 show the measured acrylamide values. Table 7 offers a statistical profile of differences in acrylamide levels for flatbreads with added fruit and vegetable peel and mushroom powders from the control values without additives.

## 4. Discussion

### 4.1. Acrylamide Content of Flatbreads, and Composition and Health Benefits of Additives

Here, we will discuss the significance of each created flatbread with the aid of the data in the tables. To stimulate interest in the newly developed “healthy” flatbreads, we briefly highlight the reported content of biologically active compounds in the additives, as well as selected health benefits for each flatbread prepared with fortified flours. Also included is a brief overview on the formation of acrylamide and related aspects.

### 4.2. Apple Peel Flatbreads

Table 4 shows that the acrylamide level of the unfortified potato flatbread of 487 µg/kg increased to 1260 µg/kg with added 5% Red Delicious apple peel, and to 906 µg/kg with 10% peel powder. By contrast, the corresponding increases in the quinoa flatbreads—from 6.4–10.5 µg/kg with 5% Red Delicious apple peel and to 11.8 µg/kg with 10%—although significant, were much lower. Surprisingly, the addition of both organic and nonorganic Fuji apple peel powders did not induce any changes in the acrylamide levels of the quinoa flatbreads (Table 5). For the whole-wheat flatbreads, addition of apple peels increased the acrylamide levels in all cases. With added Red Delicious apple peel, acrylamide increased from 29.4–52.1 µg/kg (5%) and 63.2 µg/kg (10%). The corresponding values were 46.2 µg/kg (5%) and 44.0 µg/kg (10%) for added organic Fuji apple peel, and 61.2 µg/kg (5%) and 80.5 µg/kg (10%) for nonorganic Fuji apple peel (Table 6). All of the observed increases were significant (Table 7). The mechanism underlying the increased formation of acrylamide, possibly related to the composition of reducing sugars in the different apple peels and the increased asparagine content derived from the peels, could be an area of future study. Apple peels are reported to contain triterpenoids (ursolic acid), flavonoids, and plant steroids [23,24]. These bioactive compounds seem to be responsible for their reported anti-aging [25] and anti-diabetic effects [26], and other health benefits [27]. Thus, adding apple peels to quinoa flatbreads can increase nutritional quality without generating significant amounts of acrylamide.

### 4.3. Orange Peel Flatbreads

Table 5 shows that the acrylamide content of potato flatbreads with 5% and 10% added Valencia orange peel powder, corresponding to 627 and 605 µg/kg, respectively, was higher than the value of 487 µg/kg for the flatbread without added peel. Although the increases were not dose-related, Table 7 shows that the changes were significant. The corresponding values for acrylamide content in the quinoa flatbreads of 6.4 µg/kg (no added peel), 6.9 µg/kg (with 5% peel powder), and 23 µg/kg (with 10% peel powder) showed no significant change with the addition of 5% orange peel powder, but a significant 3.6-fold increase with 10% added orange peel powder (Table 5). The corresponding three values of 29.4, 27.0, and 26.1 µg/kg for the wheat flatbreads show that the added orange peel did not affect its acrylamide levels (Table 6). The results suggest that some component of the added orange peel reacts synergistically with certain types of flours (potato, quinoa), but not others (wheat), to produce acrylamide. Since orange peels are reported to contain numerous bioactive compounds (coumarins, cinnamic acids, flavonoids) [28,29] and are reported to show anti-obesity [30], anticarcinogenic [31], and anti-diabetic [32] properties, the low-acrylamide quinoa and wheat flatbreads fortified with orange peels are expected to benefit human health.

### 4.4. Mushroom Flatbreads

The data in Table 4, Table 5, Table 6 and Table 7 show that the addition of 5 or 10% Reishi, Turkey Tail, or Lion’s Mane mushroom powders did not significantly affect the acrylamide content of the potato, quinoa, and wheat flatbreads, and could be health-promoting additives. These mushrooms are reported to contain up to 50 structurally different biologically active compounds that might be responsible for their observed antibiotic, anticancer, anti-diabetic, anti-obesity, and neuroprotective health benefits [33,34,35,36,37,38].

### 4.5. Cherry Tomato Peel Flatbreads

Table 4 shows that the acrylamide content of the potato flatbread with 5% red cherry tomato peel (691 µg/kg) was significantly greater than the potato flatbread without added peel (487 µg/kg). Surprisingly, the acrylamide content of the potato flatbread with 10% peel, or 480 µg/kg, was the same as in the flatbread without added peel. Table 5 shows that the addition of three cherry peel powders (Durst organic red, Cherub red, Zima organic orange) to the quinoa flour did not result in changes to the very low acrylamide content of the quinoa flatbread without added peels (6.4 µg/kg). Table 6 shows that the acrylamide content of the wheat flatbread of 29.4 µg/kg increased significantly to 56.0 µg/kg with 5% added organic red cherry tomato peel and to 41.8 µg/kg (not significant) with the 10% peel. For both the potato and wheat flatbreads, acrylamide increased significantly at 5% fortification, but not at the 10% level. We have no apparent explanation for these unexpected results, although there may be some antagonistic relationship between sugars and antioxidant compounds in the tomato peels, which merits further study. Tomato byproducts (peels, seeds) are reported to contain health-promoting antioxidative carotenes, lycopene, and polyphenolic compounds [39,40,41]. Unlike whole cherry tomatoes [42], cherry tomato peels have not been widely studied.

### 4.6. Potato Peel Flatbreads

Table 4 shows that adding 5% russet potato peel powder to potato flour resulted in a significant decrease in acrylamide content of the baked flatbreads from 487–367 µg/kg. Adding 10% potato peel powder resulted in a decrease to 385 µg/kg, although the result was not significant. Potato peels are reported to have numerous health benefits, including anti-obesity [12], anti-protozoan [13], and antiviral properties [43]. These properties seem to be associated with peel phenolic compounds and glycoalkaloids [44]. Whether these compounds in potato peels were responsible for reducing the formation of acrylamide can be explored in future work. However, it seems possible that adding potato peels to flatbreads may add health properties while simultaneously decreasing acrylamide levels.

### 4.7. Yam Flatbreads

Table 4 shows that the acrylamide level of 487 µg/kg in the potato flatbread increased significantly to 705 and 1150 µg/kg, respectively, with 5 and 10% added yam peel powders. There were no changes in the acrylamide levels of the corresponding quinoa flatbreads (Table 5) or in the wheat flatbreads (Table 6). As with orange peels, components of the added yam peels seem to react synergistically with only certain flours to produce acrylamide. Yams, a root and tuber crop, are a rich source of dietary fiber and could be used to create high-fiber, low-acrylamide quinoa and wheat flatbreads [45].

### 4.8. Melon Peel Flatbreads

Table 6 shows that adding 5 or 10% Galia melon peels to wheat flour did not affect the acrylamide levels of the baked flatbreads. In a previous study, we have reported that the melon peel used here contained phenolic compounds and flavonoids, and showed the highest antioxidative activity of 12 evaluated melon peel powders [16].

### 4.9. Pepino Melon Peel and Flesh Flatbreads

Table 5 shows that adding 5 or 10% pepino flesh and peel powders to quinoa flour did not change the acrylamide content of the baked flatbreads. The phenolic content of pepino (*Solanum muricatum*) is much higher than that of commercial melons [46] and is reported to inhibit inflammation and cancer by modulating the immune system [47].

## 5. Conclusions

The present study shows that the validated acrylamide analysis method had a very low 1.8 µg/kg (1.8 parts per billion) detection limit, and that adding a variety of fruit and vegetable peels and mushroom powders containing bioactive compounds to potato, quinoa, and wheat flours did not interfere with the analysis of acrylamide in the newly developed flatbreads. It was, however, surprising that the added peel powders caused increases in acrylamide content of some flatbreads by unknown mechanisms, yet had no effect in others. Since the concentration of free asparagine seems to be rate-limiting in the mechanism of formation of acrylamide, the most likely explanation for the observed increase is that some of the additives contribute free asparagine and/or reducing sugars to the flours used to prepare the flatbreads.

The finding that the addition of the three mushroom powders consistently did not enhance acrylamide levels is interesting, and indicates their potential suitability for further investigation regarding their use in creating healthier flatbreads. The levels of acrylamide in quinoa flatbreads, which were found to be low, also did not increase with the addition of most peel and mushroom powders. This finding implies that the newly-created gluten-free quinoa flatbreads had high nutritional quality [48] and potential health benefits that might be associated with the added peel and mushroom powders, and may be considered as a functional food. Although potato proteins are also of high nutritional quality [49], the peel powder-supplemented potato flatbreads experienced significant increases in acrylamide content. The variable acrylamide results with the supplemented wheat flour, whose proteins are of low nutritional quality [50], suggest that some wheat flatbreads with added peel and mushroom powders also deserve further study for their potential functionality.

The major objective of this study was to determine the acrylamide content of the flatbreads with added peels and mushroom powders, which have been reported by us and other investigators to have potential health benefits. We mention the nature of bioactive compounds reported to be present in the powders. However, based on our previous report on the effect of home heat-processing on the phenolic content of potatoes, the phenolic content of the additives may decrease during the 2 min baking of flatbreads [51]. The stability and bioactivity of phenolics in the additive powders is an important area of future research. In conclusion, the fortification of flatbreads with health-promoting supplements warrants further study, particularly for the low-acrylamide-containing flatbread with high nutritional quality and potential health benefits.

## Figures and Tables

**Figure 1 foods-08-00228-f001:**
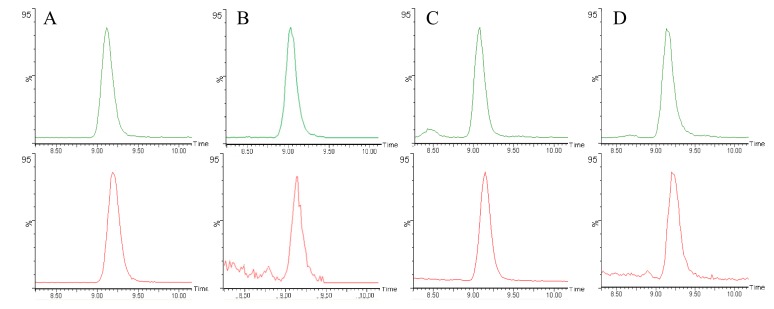
HPLC–MS/MS chromatograms of acrylamide-d3 (top) and acrylamide (bottom) in (**A**) 100 µg/L standard; (**B**) quinoa flatbread; (**C**) potato flatbread; (**D**) wheat flatbread.

**Table 1 foods-08-00228-t001:** Composition of potato flour flatbreads fortified with various fruit and vegetable peels and mushroom powders (5 or 10%).

	Additive Level	Potato Flour (g)	Salt (g)	Peel (g)	Water (mL)
Potato Flour Flatbread		75	0.75	-	125
Potato Flour Flatbread with Added:					
Organic Valencia Orange Peel	5%	75	0.75	3.75	130
	10%	75	0.75	7.50	140
Nonorganic Red Delicious Apple Peel	5%	75	0.75	3.75	130
	10%	75	0.75	7.50	140
Organic Red Cherry Tomato	5%	75	0.75	3.75	135
	10%	75	0.75	7.50	140
Organic Garnet Yam	5%	75	0.75	3.75	135
	10%	75	0.75	7.50	140
Purple Sweet Potato Yam	5%	75	0.75	3.75	130
	10%	75	0.75	7.50	135
Umatilla Russet Potato Peel	5%	75	0.75	3.75	130
	10%	75	0.75	7.50	135
Reishi Mushroom	5%	75	0.75	3.75	135
	10%	75	0.75	7.50	140
Turkey Tail Mushroom	5%	75	0.75	3.75	135
	10%	75	0.75	7.50	140
Lion’s Mane Mushroom	5%	75	0.75	3.75	135
	10%	75	0.75	7.50	140

Potato flatbreads were cooked for 2.5 min (1.25 min each side) at 195 °C. Each type of flatbread was prepared in triplicate.

**Table 2 foods-08-00228-t002:** Composition of quinoa flour flatbreads fortified with various fruit and vegetable peels and mushroom powders (5 or 10%).

	Additive Level	Quinoa Flour (g)	Salt, g	Peel, g	Water (mL)
Quinoa Flour Flatbread		100	0.80	-	100
Quinoa Flour Flatbread with Added:					
Organic Valencia Orange Peel	5%	95	0.80	5	108
	10%	90	0.80	10	112
Nonorganic Red Delicious Apple Peel	5%	95	0.80	5	104
	10%	90	0.80	10	104
Organic Fuji Apple	5%	95	0.80	5	100
	10%	90	0.80	10	100
Nonorganic Fuji Apple	5%	95	0.80	5	100
	10%	90	0.80	10	100
Organic Red Cherry Tomato Peel	5%	95	0.80	5	100
	10%	90	0.80	10	
Cherub Red Cherry Tomato Peel	5%	95	0.80	5	104
	10%	90	0.80	10	108
Zima Organic Orange Cherry Tomato Peel	5%		0.80		
	10%		0.80		
Organic Garnet Yams	5%	95	0.80	5	100
	10%	90	0.80	10	112
Purple Sweet Potato Yam Peel	5%	95	0.80	5	104
	10%	90	0.80	10	108
Pepino Melon (Egg Plant) Flesh Powder	5%	95	0.80	5	108
	10%	90	0.80	10	116
Reishi mushroom	5%	95	0.80	5	104
	10%	90	0.80	10	108
Turkey Tail mushroom	5%	95	0.80	5	104
	10%	90	0.80	10	108
Lion’s Mane mushroom	5%	95	0.80	5	104
	10%	90	0.80	10	108

Quinoa flatbreads were cooked for 2.5 min (1.25 min each side) at 195 °C. Each type of flatbread was prepared in triplicate.

**Table 3 foods-08-00228-t003:** Composition of whole-wheat flour flatbreads fortified with various fruit and vegetable peels and mushroom powders (5 or 10%).

	Additive Level	Wheat Flour (g)	Salt (g)	Peel (g)	Water (mL)
Whole-Wheat Flour Flatbread		100	0.80	-	90
Wheat Flour Flatbread with Added:					
Organic Valencia Orange Peel	5%	100	0.84	5	94.5
	10%	100	0.88	10	99
Organic Naval Orange Peel	5%	100	0.84	5	94.5
	10%	100	0.88	10	99
Nonorganic Mandarin Orange Peel	5%	100	0.84	5	94.5
	10%	100	0.88	10	99
Nonorganic Red Delicious Apple Peel	5%	100	0.84	5	94.5
	10%	100	0.88	10	99
Organic Fuji Apple Peel	5%	100	0.84	5	94.5
	10%	100	0.88	10	99
Nonorganic Fuji Apple	5%	100	0.84	5	94.5
	10%	100	0.88	10	99
Organic Red Cherry Tomato Peel	5%	100	0.84	5	94.5
	10%	100	0.88	10	99
Organic Garnet Yams	5%	100	0.84	5	94.5
	10%	100	0.88	10	99
Purple Sweet Potato Yams	5%	100	0.84	5	94.5
	10%	100	0.88	10	99
Pepino Melon Flesh Powder	5%	100	0.84	5	94.5
	10%	100	0.88	10	99
Organic Galia Melon	5%	100	0.84	5	94.5
	10%	100	0.88	10	99
Nonorganic Galia Melon	5%	100	0.84	5	94.5
	10%	100	0.88	10	99
Reishi mushroom	5%	100	0.84	5	94.5
	10%	100	0.88	10	99
Turkey Tail mushroom	5%	100	0.84	5	94.5
	10%	100	0.88	10	99
Lion’s Mane mushroom	5%	100	0.84	5	94.5
	10%	100	0.88	10	99

Whole-wheat flatbreads were cooked for 2 min (1 min each side) at 195 °C. Each flatbread was prepared in triplicate.

**Table 4 foods-08-00228-t004:** Effect of 5 and 10% peel powders added to potato flour on the acrylamide content of flatbreads after baking at 195 °C for 2.5 min.

Flatbread	Additive Amount	Acrylamide (µg/kg)
Potato	0%	487 ± 39
Potato with Added:		
Valencia Orange Peel	5%	627 ± 58 *
	10%	605 ± 26 *
Red Delicious Apple Peel	5%	1260 ± 162 *
	10%	906 ± 89 *
Organic Red Cherry Tomato Peel	5%	691 ± 47 *
	10%	480 ± 11
Organic Garnet Sweet Potato Yam	5%	716 ± 60 *
	10%	716 ± 79 *
Purple Sweet Potato Yam	5%	705 ± 88 *
	10%	1150 ± 152 *
Umatilla Russet Potato Peel	5%	367 ± 12 *
	10%	385 ± 50
Reishi Mushroom	5%	518 ± 60
	10%	380 ± 27 *
Turkey Tail Mushroom	5%	570 ± 40
	10%	472 ± 26
Lion’s Mane Mushroom	5%	440 ± 75
	10%	452 ± 70

* Treatment flatbreads were significantly different from the control (no additive) using a two-tailed independent samples test (*p* ≤ 0.05).

**Table 5 foods-08-00228-t005:** Effect of 5 and 10% peel powders added to quinoa flour on the acrylamide content of flatbreads after baking at 195 °C for 2.5 min.

Flatbread	Additive Amount	Acrylamide (µg/kg)
Quinoa (no additives)	0%	6.4 ± 1.3
Quinoa with Added:		
Valencia Orange Peel	5%	6.9 ± 0.9
	10%	23.0 ± 7.6 *
Red delicious apple peel	5%	10.5 ± 1.4 *
	10%	11.8 ± 1.2 *
Organic Fuji Apple Peel	5%	4.4 ± 0.3 ^1^
	10%	4.9 ± 0.1 ^1^
Nonorganic Fuji Apple Peel	5%	6.4 ± 2.0
	10%	7.4 ± 1.1
Durst Organic Red Cherry Tomato Peel	5%	4.2 ± 0.6 ^1^
	10%	4.5 ± 0.8 ^1^
Cherub Red Cherry Tomato Peel	5%	6.4 ± 0.5
	10%	5.7 ± 0.1 ^1^
Zima Organic Orange Cherry Tomato Peel	5%	4.9 ± 0.8 ^1^
	10%	4.7 ± 1.4 ^1^
Organic Garnet Yams	5%	4.1 ± 1.0 ^1^
	10%	5.3 ± 1.0 ^1^
Purple Sweet Potato Yams	5%	4.3 ± 0.5 ^1^
	10%	6.8 ± 1.2
Pepino Melon Flesh	5%	3.5 ± 0.6 ^1,^*
	10%	3.8 ± 0.2 ^1,^*
Pepino Melon Peel	5%	4.5 ± 0.5 ^1^
	10%	3.8 ± 0.4 ^1,^*
Lion’s Mane Mushroom	5%	4.3 ± 0.5 ^1^
	10%	5.6 ± 0.2 ^1^
Reishi Mushroom	5%	7.2 ± 0.8
	10%	6.7 ± 0.5
Turkey Tail Mushroom	5%	5.0 ± 1.0 ^1^
	10%	5.2 ± 1.0 ^1^

^1^ Acrylamide concentrations were below the limit of quantification (6 µg/kg), but estimated values are shown. * Treatment flatbreads were significantly different from the control (no additive) using a two-tailed independent samples test (*p* ≤ 0.05).

**Table 6 foods-08-00228-t006:** Effect of 5 and 10% peel powders added to whole-wheat flour on the acrylamide content of flatbreads after baking at 195 °C for 2.0 min.

Flatbread	Additive Amount	Acrylamide (µg/kg)
Whole Wheat	0%	29.4 ± 6.5
Whole Wheat with Added:		
Valencia Orange Peel	5%	27.0 ± 4.1
	10%	26.1 ± 2.1
Navel Orange Peel (Organic)	5%	23.9 ± 2.3
	10%	22.9 ± 4.1
Mandarin Orange Peel (Nonorganic)	5%	20.6 ± 1.5
	10%	23.3 ± 0.6
Red Delicious Apple Peel	5%	52.1 ± 5.4 *
	10%	63.2 ± 8.0*
Fuji Apple Peel (Organic)	5%	46.2 ± 4.8 *
	10%	44.0 ± 4.5*
Fuji Apple Peel (Nonorganic)	5%	61.2 ± 8.1 *
	10%	80.5 ± 9.9 *
Red Cherry Tomato Peel (Organic)	5%	56.0 ± 7.7 *
	10%	41.8 ± 7.0
Garnet Yam (Organic)	5%	37.3 ± 3.0
	10%	30.6 ± 1.4
Purple Sweet Potato Yam	5%	32.2 ± 4.2
	10%	34.8 ± 5.5
Pepino Melon Flesh	5%	57.2 ± 7.5 *
	10%	58.3 ± 17
Galia Melon (Organic)	5%	30.8 ± 3.0
	10%	37.3 ± 4.2
Galia Melon (Nonorganic)	5%	26.9 ± 1.9
	10%	25.7 ± 3.4
Reishi Mushroom	5%	40.9 ± 2.7
	10%	35.6 ± 3.6
Turkey Tail Mushroom	5%	27.8 ± 3.4
	10%	25.2 ± 3.0
Lion’s Mane Mushroom	5%	40.7 ± 6.2
	10%	33.8 ± 1.4

* Treatment flatbreads were significantly different from the control (no additive) using a two-tailed independent samples test (*p* ≤ 0.05).

**Table 7 foods-08-00228-t007:** Summary of the effects of fruit and vegetable peels and mushroom powder additives on acrylamide content.

	Potato		Quinoa		Whole Wheat	
Additive Fortified at	5%	10%	5%	10%	5%	10%
Valencia orange	+	+	0	+ +	+	+
Navel orange					0	0
Mandarin orange					0	0
Red delicious apple	+ +	+ +	+ +	+ +	+ +	+ +
Fuji apple (organic)			0	0	+ +	+ +
Fuji apple (nonorganic)			0	0	+ +	+ +
Durst red cherry tomato	+	0	0	0	+ +	0
Garnet yam	+	+	0	0	0	0
Purple sweet potato	+	+ +	0	0	0	0
Umatilla Russet potato	–	0				
Pepino melon flesh			–	–	+ +	+ +
Pepino melon peel			0	–		
Galia melon					0	0
Reishi mushroom	0	–	0	0	0	0
Lion’s Mane mushroom	0	0	0	0	0	0
Turkey Tail mushroom	0	0	0	0	0	0

Significant increases or decreases from control (no additive) were determined using a two-tailed independent samples test (*p* ≤ 0.05). (–) = 0–50% decrease; 0 = no effect; (+) = 0–50% increase; (+ +) = <50% increase. Blank means the additive was not tested in that matrix.

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
