# Peer review of "Acrylamide Content of Experimental Flatbreads Prepared from Potato, Quinoa, and Wheat Flours with Added Fruit and Vegetable Peels and Mushroom Powders"

_foods, 2019, doi:10.3390/foods8070228_

Reviewer 1 Report

The topic is of current interest and relevant for both consumers and food industries, however, I believe some aspects need to be deepened or clarified to improve the quality of paper before publication.

- It would be appropriate to include in the introduction, a brief overview regarding the process of formation of acrylamide, the adverse effects on nutrition and the measures that can be taken to reduce its content/formation, as well as the dose ranges specified by regulations, which can help the reader to contextualize the effective impact on nutrition of the specific food intake.  

- Why was the amount of water used to prepare the doughs different for each thesis within the main ingredient? Is it possible that the different dough yield, although minimal, influenced acrylamide formation during the cooking process?

- I found the argument of the results rather meagre. The authors set the entire discussion around the potential benefits of the powders tested and attributed the differences within and among the matrices to their different composition and the possible interaction with other compounds. Yet no characterization of the powders was performed. How did the authors explain the increase in acrylamide concentration when it was not dose-related? I understand analyzing the chemical profile of all matrices is unlikely to be performed, therefore, my suggestion is to extensively review the literature and collecting as many data as possible (eg. sugars and amino acids composition), to add more concrete hypothesis supporting your results.

-The substantial citation of papers claiming the functionality of the matrices employed is borderline speculation. Not only many of the functional properties citated are associated to phenolic compounds which can be drastically impacted by the cooking treatment, but also the bioactivity of these compounds needs to be demonstrated by in vitro and in vivo studies. It would have been appropriate, after the first screening based on the acrylamide content, to select the best thesis and characterize them for the features of nutritional interest.

-Since food appearances and taste are the major drivers of consumers choice, have the authors considered to perform a sensorial analysis of the thesis which gave better results?

Line 105. I assume “fortified” is missing from the description of the table

Line 153. The letters in the figure cannot be properly seen

Author Response

Reviewer 1:

Comments and Suggestions for Authors

The topic is of current interest and relevant for both consumers and food industries, however, I believe some aspects need to be deepened or clarified to improve the quality of paper before publication.

It would be appropriate to include in the introduction, a brief overview regarding the process of formation of acrylamide, the adverse effects on nutrition and the measures that can be taken to reduce its content/formation, as well as the dose ranges specified by regulations, which can help the reader to contextualize the effective impact on nutrition of the specific food intake.

Response: Suggestion adopted.  We added a brief overview on the formation and adverse effects of acrylamide and related aspects as suggested by the Reviewer.  Guidelines for maximum acrylamide levels in food recommended by different government agencies and the estimated consumption of acrylamide by different populations in 14 countries are described in our previous publication (reference 8). 

See page 1, lines 30-43.

- Why was the amount of water used to prepare the doughs different for each thesis within the main ingredient? Is it possible that the different dough yield, although minimal, influenced acrylamide formation during the cooking process

:ResponseTalwinder: To make optimized dough with different flours and ingredients the amount of water needed varies due difference in rheological properties of the ingredients.  We do not know if these small differences in the amount of water needed would influence the acrylamide production.

- I found the argument of the results rather meagre. The authors set the entire discussion around the potential benefits of the powders tested and attributed the differences within and among the matrices to their different composition and the possible interaction with other compounds. Yet no characterization of the powders was performed. How did the authors explain the increase in acrylamide concentration when it was not dose-related? I understand analyzing the chemical profile of all matrices is unlikely to be performed, therefore, my suggestion is to extensively review the literature and collecting as many data as possible (eg. sugars and amino acids composition), to add more concrete hypothesis supporting your results.

Response:  The major objective of this study was to determine the acrylamide content of the flatbreads with added fruit and antioxidative vegetable peels and mushroom powders that have been reported by us and other investigators to have potential health benefits.  In each case, we mention the nature of bioactive compounds reported to be present in the powders. 

Based on our previous report (ref 51) on the effect home heat processing on the phenolic content of potatoes, we agree with the reviewer that there might be a decrease in phenolic content during the 2-min baking of the flatbreads.

Because the concentration of free asparagine seems to be rate-limiting in the mechanism of formation of acrylamide, the most likely explanation for the observed increase is that some of the additives contribute free asparagine and/or reducing sugars to the flours used to prepare the flatbreads.

These aspects are now mentioned in the Conclusion section on page 14.

-The substantial citation of papers claiming the functionality of the matrices employed is borderline speculation. Not only many of the functional properties citated are associated to phenolic compounds which can be drastically impacted by the cooking treatment, but also the bioactivity of these compounds needs to be demonstrated by in vitro and in vivo studies. It would have been appropriate, after the first screening based on the acrylamide content, to select the best thesis and characterize them for the features of nutritional interest.

-Since food appearances and taste are the major drivers of consumers choice, have the authors considered to perform a sensorial analysis of the thesis which gave better results?

We expect that this publication will stimulate interest in sensory studies of the flatbreads for commercial use. 

Line 105. I assume “fortified” is missing from the description of the table.  Corrected.

Line 153. The letters in the figure cannot be properly seen.

The letters in Figure 1 were enlarged.

Reviewer 2 Report

Presented article seems to be interesting and concerns the acrylamide content in different types of flatbread.  I have only a few shortcomings which should be taken into consideration:  

Line 92, It is inappropriate to give the temperature in the manuscript in the different units °F or °C (line 94, 97).

Line 93: bread was cooked or baked?

Lines 131-135 What was the level of significance for evaluation the statistical differences between means?

Lines 138-140: This information should be included in chapter “Materials and methods”

Lines 182-183 and in the whole manuscript: It is inappropriate to give values of acrylamide level without units. It should be changed in the whole manuscript.   

Author Response

Comments and Suggestions for Authors

Presented article seems to be interesting and concerns the acrylamide content in different types of flatbread.  I have only a few shortcomings which should be taken into consideration:  

Line 92, It is inappropriate to give the temperature in the manuscript in the different units °F or °C (line 94, 97).

Response: All units are now in oC.

Line 93: bread was cooked or baked?

Response. Explanation now included. Flatbreads were cooked between upper and lower hot irons of the flatbread maker. See lines 97-98.

Lines 131-135 What was the level of significance for evaluation the statistical differences between means?

revised Figure 1 and added references and units for acrylamide values in the discussion. I also included p-values in response to the statistics question (line 127).

We added (p ≤ 0.05) to the section on Statistics. 

Lines 138-140: This information should be included in chapter “Materials and methods”

We moved the two sentences from Results to Material and methods, as suggested by the reviewer.

Lines 182-183 and in the whole manuscript: It is inappropriate to give values of acrylamide level without units. It should be changed in the whole manuscript.

Units were added to all acrylamide values, as suggested by the reviewer.

Round  2

Reviewer 1 Report

Suggested changes improved the quality of the manuscript.